# Early use of barbiturates is associated with increased mortality in traumatic brain injury patients from a propensity score-based analysis of a prospective cohort

Maxime Léger[1,2]*, Denis Frasca[2,3], Antoine Roquilly[4], Philippe Seguin[5], Raphaël Cinotti[4], Claire Dahyot-Fizelier[3], Karim Asehnoune[4], Florent Le Borgne[2,6], Thomas Gaillard[1], Yohann Foucher[2,7], Sigismond Lasocki[1], for AtlanRéa group[¶]

1 Département d'Anesthésie Réanimation, Centre Hospitalier Universitaire d'Angers, Angers, France, 2 INSERM UMR 1246—SPHERE, Nantes University, Tours University, Nantes, France, 3 Département d'Anesthésie Réanimation, Centre Hospitalier Universitaire de Poitiers, Poitiers, France, 4 Département d'Anesthésie Réanimation, Centre Hospitalier Universitaire de Nantes, Nantes, France, 5 Département d'Anesthésie Réanimation, Centre Hospitalier Universitaire de Rennes, Rennes, France, 6 IDBC-A2COM, Pacé, France, 7 Centre Hospitalier Universitaire de Nantes, Nantes, France

¶ The complete membership of the author group can be found in the Acknowledgments
* maxime.leger@chu-angers.fr

**Data Availability Statement:** All relevant data are within the paper and its Supporting Information files.

## Abstract

Barbiturates are proposed as a second/third line treatment for intracranial hypertension in traumatic brain injury (TBI) patients, but the literature remains uncertain regarding their benefit/risk balance. We aimed to evaluate the impact of barbiturates therapy in TBI patients with early intracranial hypertension on the intensive care unit (ICU) survival, the occurrence of ventilator-associated pneumonia (VAP), and the patient's functional status at three months. We used the French AtlanREA prospective cohort of trauma patients. Using a propensity score-based methodology (inverse probability of treatment weighting), we compared patients having received barbiturates within the first 24 hours of admission (barbiturates group) and those who did not (control group). We used cause-specific Cox models for ICU survival and risk of VAP, and logistic regression for the 3-month Glasgow Outcome Scale (GOS) evaluation. Among the 1396 patients with severe trauma, 383 had intracranial hypertension on admission and were analyzed. Among them, 96 (25.1%) received barbiturates. The early use of barbiturates was significantly associated with increased ICU mortality (HR = 1.85, 95%CI 1.03–3.33). However, barbiturates treatment was not significantly associated with VAP (HR = 1.02, 95%CI 0.75–1.41) or 3-month GOS (OR = 1.67, 95%CI 0.84–3.33). Regarding the absence of relevant clinical trials, our results suggest that each early prescription of barbiturates requires a careful assessment of the benefit/risk ratio.

**Funding:** The authors received no specific funding for this work.

**Competing interests:** The authors have declared that no competing interests exist.

## Introduction

Monitoring intracranial pressure (ICP) is recommended for the care of patients with severe traumatic brain injury (TBI) [1]. Indeed, even in brief periods, intracranial hypertension is associated with poorer outcomes [2–4]. Many therapeutic options are available in case of elevated ICP, including sedation, osmotherapy, maintenance of high cerebral perfusion pressures, external ventricular drainage, craniectomy, etc. A stepwise implementation of these treatments is usually proposed, and barbiturates are one of these options [1].

Indeed, barbiturates have been recommended to treat high and refractory ICP since the early 80s [5,6]. They are still suggested as a second or third line of treatment in US guidelines [7], or in the recent Seattle International Severe Traumatic Brain Injury Consensus Conference [1], when the increase in ICP is refractory to other medical or surgical alternatives in TBI patients. Nonetheless, the body of evidence for the use of barbiturates in this context is still of low quality. Since the first randomized clinical trial by Eisenberg et al. in 1988, evaluating high doses of barbiturates for intractable ICP elevation in patients with a Glasgow Coma Scale (GCS) of 4–8 [8], few data are available. The most recent Cochrane review, including only seven trials with a total of 341 patients, concluded that there was no evidence for the use of barbiturates therapy in patients with an acute severe head injury [9]. In this analysis, there was no significant effect of barbiturates on mortality or recovery at the end of follow-up.

This lack of proven efficacy is even more worrying in the light of the known adverse effects of barbiturates. Beyond the induced hypotension described since the first uses of this therapy [10], barbiturates could induce immunosuppression [11], promoting the occurrence of ventilator-associated pneumonia (VAP) [12,13]. However, nowadays, barbiturates appear to be still widely used in TBI patients [14].

In this context of widespread use and absence of a well-established benefit-risk balance, further data is needed. The purpose of this study was to evaluate in real-life settings the impact of early barbiturates infusion on death, neurological outcome, and occurrence of VAP in patients admitted to Intensive Care Units (ICU) with severe TBI and intracranial hypertension.

## Materials and methods

### Study population

We used the prospective multicentric AtlanREA cohort (www.atlanrea.org) of trauma patients (NCT02426255), hospitalized in ICU in the western region of France. Data confidentiality was ensured following the recommendations of the French commission for data protection (Commission Nationale Informatique et Liberté, CNIL decisions DR-2013-047). The study was approved by the Ethics Committee of the French society of anesthesia and intensive care (SFAR), which waived patients' consent for this study according to French law on Bioethics [15]. All patients or their relatives were informed and agreed to the data collection. For this study, all methods were performed in accordance with relevant guidelines and regulations.

All patients in the database aged 15 years or more and hospitalized between March 2013 and March 2018 in one of the four university hospitals (Angers, Nantes, Poitiers, and Rennes) were eligible if they had TBI (defined as a brain lesion on the initial CT-scan) requiring admission to ICU with orotracheal intubation and had intracranial hypertension within the first 24 hours of admission. Intracranial hypertension was defined as an ICP >20 mmHg [16], or the need for a second/third-line therapy for clinical intracranial hypertension (i.e., osmotherapy, infusion of barbiturates, external ventricular drainage, urgent neurosurgery) because of the presence of a focal sign on admission before ICP monitoring.

Patients who received barbiturates (at least a bolus of 250 mg) within the first 24-hours of admission were considered in the barbiturates group, while those who did not were in the control group. The barbiturates group thus only included early administration of barbiturates (within the first 24 hours), in order to build a relevant propensity score with the maximum of variables (we did not have all the data later during the ICU stay).

## Available data

Patient characteristics recorded were age, gender, Body Mass Index (BMI), admission Simplified Acute Physiology Score II (SAPS II) [17], worst GCS score (detailing the GCS motor score) during the first 24 hours following ICU admission, and the Injury Severity Score (ISS) [18]. Episodes of unreactive anisocoria or mydriasis, hypoxia, low blood pressure (defined as a systolic blood pressure < 90 mmHg), or blood transfusion were also noted before admission (i.e., during transportation, in the emergency room or the operating room). We also collected comorbidities, including histories of heart failure, renal failure, chronic respiratory disease, diabetes, cancer, and chronic tobacco- or alcohol use. Biological parameters available at ICU admission were hemoglobin, leukocytes, prothrombin time, platelets, fibrinogen, arterial lactate, pH, bicarbonate, the partial pressure of oxygen (PaO2), the fraction of inspired oxygen (fiO2), serum creatinine, protein, glucose, urea, and calcium. Computed tomography severity was specified using Marshall's classification (from I to VI, Class I being indicative for less severe lesions) [19].

The use of other specific neurocritical care therapeutics was also collected: osmotherapy, surgical intracerebral hematoma evacuation, external ventricular drainage, lobectomy, or decompressive craniectomy.

## Outcomes

The primary outcome was patient survival in ICU. We also studied the time-to-first VAP, defined according to a standardized definition [20], and the 3-month post-admission Glasgow Outcome Scale (GOS) [21], dichotomized into favorable outcome (good recovery or moderate disability) or unfavorable outcome (severe disability, vegetative state or death).

## Statistical analysis

We compared the patient's characteristics between the two groups of interest using Chi-square tests for categorical variables and Student t-tests for continuous variables. For outcomes' comparisons, we weighted on the Propensity Scores (PS) to consider possible confounders. The PS was estimated by multivariable logistic regression. Splines on continuous covariates were used to ensure the log-linearity assumptions. Variables significantly associated with the outcome and treatment in univariate regressions were retained (p<0.2). In addition, we also studied propensity score models including less variables by using the IMPACT TBI score (composed of the following covariates: age, Glasgow motor score, pupillary reactivity, hypoxia and hypotension status, and Marshall's CT scan score) since this score is a good predictor of TBI severity [22]. We removed in these models the variables already included in the IMPACT TBI score (i.e., SAP ≤ 90 mmHg, and the diagnosis of extradural or subdural hematoma), to avoid collinearities. These sensitivity analyses were carried out for the ICU survival and the 3-month GOS. For all the analyses, we considered a center effect as a covariate in the PS. We applied stabilized weights estimating the average treatment effect in the entire population (ATE) [23,24]. We assessed the goodness-of-fit of the models by checking the positivity assumptions graphically and studying standardized differences.

For times-to-event, cause-specific Cox models were estimated by maximizing the partial weighted likelihood and using a robust estimator for the variance [25]. Hazard

proportionalities were graphically checked by plotting log-minus-log survival curves. The crude cumulative incidence curves were obtained by the Aalen-Johansen estimator to account for competing risks. For the three-month GOS, we used a logistic regression by maximizing the weighted likelihood and using a robust estimator for the variance. Influential values were detected by a Cook distance greater than one in absolute value.

### Sensitivity analysis

As stated above, we had to focus our study on the early administration of barbiturates to be able to build a relevant propensity score. However, some patients received barbiturates later during their stay. We conducted a sensitivity analysis by comparing ICU survival among patients who received barbiturates infusion at any time during the stay to those who did not. We used the same statistical approach (i.e., a cause-specific cox model). Such an analysis did not consider the immortal time bias and should be interpreted with caution.

All the statistical analyses were performed using the Plug-Stat software (www.labcom-risca. com) based on the R software (R Core Team, 2017, version 3.4.0). All the candidate variables for calculating propensity scores are those presented in **Table 1** (**S1 Tables** in S1 File detail multivariable logistic models leading to the propensity scores, **S2 Table** in S1 File show the standardized differences, and **S1 Fig** in S1 File resume the propensity scores distributions).

## Results

### Cohort description

During the study period, 1396 trauma patients were included in the database. Among them 982 patients had severe TBI and 699 required the insertion of an intracranial pressure monitoring sensor. 383 TBI patients had an intracranial hypertension (i.e., ICP >20 mmHg [16] or the need for therapy of intracranial hypertension because of localizing signs on admission) within the first 24-hours of admission and were included in the analysis, as depicted in **Fig 1**. Ninety-six (25.1%) patients were treated with barbiturates within the first 24 hours of admission, resulting in a control group of 287 (74.9%) patients. As illustrated in **Table 1**, patients treated with barbiturates tended to be younger and had a higher proportion of unreactive mydriasis or anisocoria before admission, GCS scores, but the CT severity of the injuries did not differ between the two groups. The proportion of blood transfusion before admission was higher in the barbiturates group, and these patients received more osmotherapy and had higher ICP at admission. We identified additional differences in plasma fibrinogen, plasma bicarbonate, plasma glucose, plasma proteins, and plasma calcium levels between groups. The IMPACT TBI score was close between the two groups, with a mean value of 9.6 in the barbiturates group versus 8.5 in the control group. According to these values, the calculated probability of death at 6 months was 41.8% in the barbiturates group versus 34.6% in the control group (p = 0.5121).

### Patient survival in ICU

During the follow-up, 117 (30.5%) patients have died while in the ICU, including 44 (45.8%) patients in the barbiturates group and 73 (25.4%) patients in the control group. The cumulative probabilities of death in the ICU are presented in **Fig 2**. The observed (non-adjusted) cause-specific HR of death in the ICU was 2.13 (95%CI from 1.45 to 3.13) for patients of the barbiturates group versus those of the control group. After weighting on propensity scores, the corresponding confounder-adjusted HR was 1.85 (95%CI from 1.03 to 3.33). When we considered the propensity score model which included the IMPACT TBI score, the confounder-adjusted HR was 1.85 (95%CI from 1.04 to 3.23).

**Table 1.  Description of the studied population at baseline.**

| | Overall (n = 383) | | | Control group (n = 287) | | | Barbiturates group (n = 96) | | | p-value |
|---|---|---|---|---|---|---|---|---|---|---|
| | NA | n | % | NA | n | % | NA | n | % | |
| **Male** | 0 | 308 | 80.4 | 0 | 230 | 80.1 | 0 | 78 | 81.2 | 0.8124 |
| **History of diabetes** | 5 | 22 | 5.8 | 4 | 16 | 5.7 | 1 | 6 | 6.3 | 0.8115 |
| **Chronic alcoholism** | 31 | 64 | 18.2 | 24 | 53 | 20.2 | 7 | 11 | 12.4 | 0.0994 |
| **Active smoking status** | 61 | 94 | 29.2 | 49 | 65 | 27.3 | 12 | 29 | 34.5 | 0.2113 |
| **SAP $\leq$ 90 mmHg** | 7 | 105 | 27.9 | 6 | 71 | 25.3 | 1 | 34 | 35.8 | 0.0481 |
| **Blood transfusion** | 1 | 99 | 25.9 | 1 | 65 | 22.7 | 0 | 34 | 35.4 | 0.0141 |
| **Hypoxia** | 10 | 279 | 74.8 | 5 | 206 | 73.0 | 5 | 73 | 80.2 | 0.2183 |
| **Glasgow score $<$ 8** | 5 | 104 | 26.2 | 1 | 74 | 25.5 | 4 | 30 | 28.3 | 0.6036 |
| **Glasgow motor score** | 20 | | | 12 | | | 8 | | | 0.3569 |
| 6 | | 35 | 9.6 | | 27 | 9.8 | | 8 | 9.1 | |
| 5 | | 63 | 17.4 | | 51 | 18.5 | | 12 | 13.6 | |
| 4 | | 82 | 22.6 | | 66 | 24.0 | | 16 | 18.2 | |
| 3 | | 36 | 9.9 | | 27 | 9.8 | | 9 | 10.2 | |
| 2 | | 34 | 9.4 | | 23 | 8.4 | | 11 | 12.5 | |
| 1 | | 113 | 31.1 | | 81 | 29.5 | | 32 | 36.4 | |
| **Unreactive mydriasis or anisocoria** | 5 | 140 | 36.6 | 5 | 96 | 33.4 | 0 | 44 | 45.8 | 0.0498 |
| **CT scan classification** **Marshall classification** | 0 | | | 0 | | | 0 | | | 0.2550 |
| Diffuse injury I | | 10 | 2.6 | | 10 | 3.5 | | 0 | 0.0 | |
| Diffuse injury II | | 89 | 23.2 | | 72 | 25.1 | | 17 | 17.7 | |
| Diffuse injury III | | 33 | 8.6 | | 23 | 8.0 | | 10 | 10.4 | |
| Diffuse injury IV | | 25 | 6.5 | | 18 | 6.3 | | 7 | 7.3 | |
| Evacuated mass lesion V | | 151 | 39.5 | | 108 | 37.6 | | 43 | 44.8 | |
| Non-evacuated mass VI | | 75 | 19.6 | | 56 | 19.5 | | 19 | 19.8 | |
| **Osmotherapy** | 2 | 255 | 66.9 | 1 | 177 | 61.9 | 1 | 78 | 82.1 | 0.0003 |
| **Evacuation of subdural or extradural hematoma** | 0 | 108 | 28.2 | 0 | 74 | 25.8 | 0 | 34 | 35.4 | 0.0694 |
| **External ventricular drain** | 0 | 28 | 7.3 | 0 | 22 | 7.7 | 0 | 6 | 6.2 | 0.6447 |
| **Evacuation of cerebral hematoma** | 0 | 17 | 4.4 | 0 | 10 | 3.5 | 0 | 7 | 7.3 | 0.1493 |
| **Decompressive craniectomy** | 0 | 74 | 19.3 | 0 | 51 | 17.8 | 0 | 23 | 24.0 | 0.1837 |
| | NA | mean | SD | NA | mean | SD | NA | mean | SD | |
| **Age (years)** | 0 | 40.5 | 18.7 | 0 | 41.7 | 19.2 | 0 | 36.9 | 16.8 | 0.0194 |
| **BMI (kg.m-2)** | 48 | 24.3 | 4.5 | 32 | 24.4 | 4.5 | 16 | 23.8 | 4.8 | 0.3545 |
| **Intracranial pressure on admission (mm Hg)** | 17 | 22.0 | 15.8 | 14 | 19.3 | 13.3 | 3 | 29.8 | 19.5 | 0.0001 |
| **Haemoglobin (g/dL)** | 3 | 11.0 | 2.5 | 1 | 11.1 | 2.4 | 2 | 10.8 | 2.6 | 0.3831 |
| **Leukocytes (count/mm$^3$)** | 4 | 17.9 | 7.1 | 2 | 17.5 | 7.1 | 2 | 18.9 | 7.0 | 0.0909 |
| **Prothrombin (%)** | 15 | 69.6 | 18.9 | 12 | 70.4 | 18.5 | 3 | 67.4 | 20.2 | 0.2120 |
| **Platelets (count/mm$^3$)** | 4 | 178.4 | 70.0 | 2 | 180.5 | 66.8 | 2 | 172.0 | 79.0 | 0.3489 |
| **Fibrinogen (g/L)** | 70 | 2.3 | 1.2 | 53 | 2.4 | 1.2 | 17 | 2.1 | 1.3 | 0.0582 |
| **Arterial Lactate (mmol/L)** | 52 | 2.8 | 2.3 | 38 | 2.7 | 2.3 | 14 | 3.1 | 2.2 | 0.1055 |
| **Arterial pH** | 4 | 7.3 | 0.1 | 2 | 7.3 | 0.1 | 2 | 7.3 | 0.1 | 0.0659 |
| **Bicarbonate (mmol/L)** | 4 | 20.9 | 4.1 | 2 | 21.2 | 4.0 | 2 | 20.0 | 4.4 | 0.0175 |
| **PaO2 (mm Hg)** | 8 | 136.6 | 79.2 | 5 | 134.7 | 75.9 | 3 | 142.6 | 88.5 | 0.4437 |
| **FiO2** | 9 | 0.5 | 0.2 | 4 | 0.5 | 0.2 | 5 | 0.5 | 0.2 | 0.7814 |
| **PaO2/FiO2 ratio** | 14 | 321.0 | 179.4 | 8 | 323.6 | 190.8 | 6 | 313.0 | 138.7 | 0.5693 |
| **Serum Creatinine (mmol/L)** | 4 | 78.5 | 31.4 | 3 | 76.6 | 31.0 | 1 | 84.4 | 32.1 | 0.0395 |
| **Serum proteins (g/L)** | 9 | 54.3 | 11.0 | 6 | 54.9 | 11.1 | 3 | 52.4 | 10.8 | 0.0585 |
| **Serum glucose (mmol/L)** | 31 | 8.7 | 3.4 | 17 | 8.5 | 2.9 | 14 | 9.6 | 4.7 | 0.0352 |

*(Continued)*

**Table 1.** (Continued)

| | Overall (n = 383) | | | Control group (n = 287) | | | Barbiturates group (n = 96) | | | p-value |
|---|---|---|---|---|---|---|---|---|---|---|
| | NA | n | % | NA | n | % | NA | n | % | |
| **Serum urea (mmol/L)** | 9 | 4.9 | 2.1 | 8 | 4.8 | 2.0 | 1 | 5.0 | 2.3 | 0.6403 |
| **Serum calcium (mmol/L)** | 42 | 2.0 | 0.2 | 32 | 2.0 | 0.2 | 10 | 1.9 | 0.2 | 0.1091 |
| **SAPS II score** | 19 | 46.0 | 12.4 | 13 | 45.3 | 11.9 | 6 | 47.9 | 13.6 | 0.1091 |
| **ISS score** | 2 | 27.1 | 14.7 | 1 | 27.1 | 14.5 | 1 | 27.1 | 15.2 | 0.9842 |
| **IMPACT TBI score** | 38 | 8.8 | 4.3 | 11 | 8.5 | 4.2 | 27 | 9.6 | 4.4 | 0.0654 |

NA, number of data Not Available; BMI, Body Mass Index; FIO2, Fraction of Inspired Oxygen; ISS score, Injury Severity Score; PaO2, arterial partial Pressure of Oxygen; SAP, Systolic Arterial Pressure; SAPS, Simplified Acute Physiology Score; SD, standard deviation. IMPACT TBI score includes: age, Glasgow motor score, pupillary reactivity, hypoxia and hypotension status, and Marshall's CT scan score and is predictive of mortality in TBI patients.

## Incidence of VAP

Seven patients treated for aspiration pneumonia on admission in the ICU were excluded. Among the remaining 376 patients, 207 (55.1%) developed a least one VAP during their ICU

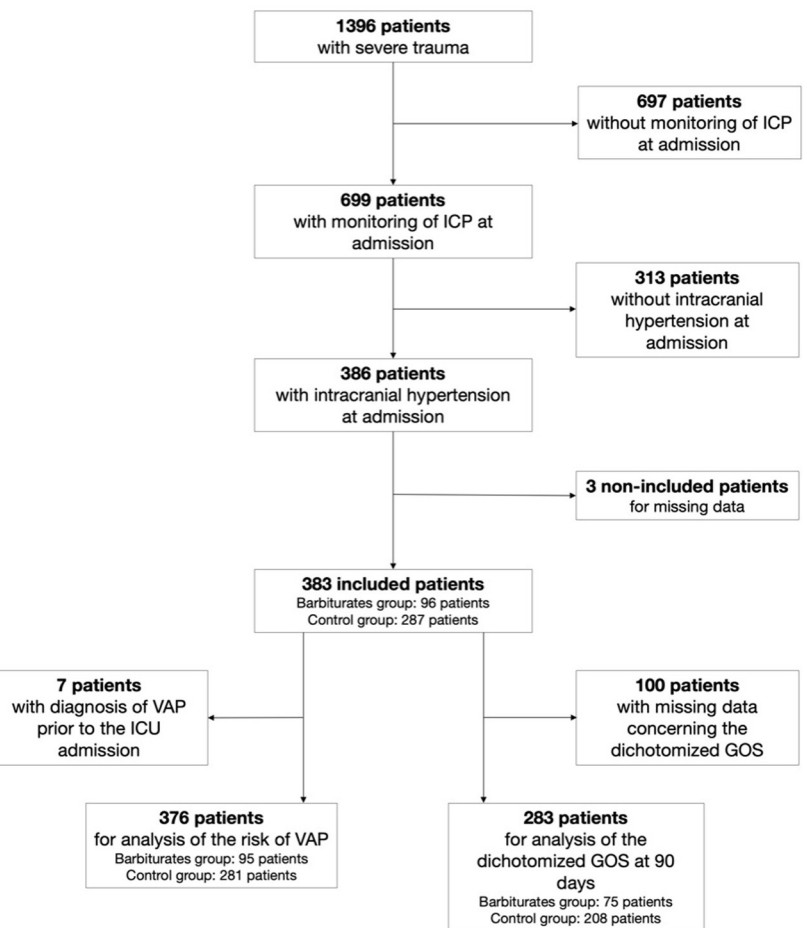

**Fig 1. Flow chart of TBI patients of the AtlanREA cohort who had intracranial pressure monitoring and developed intracranial hypertension.**

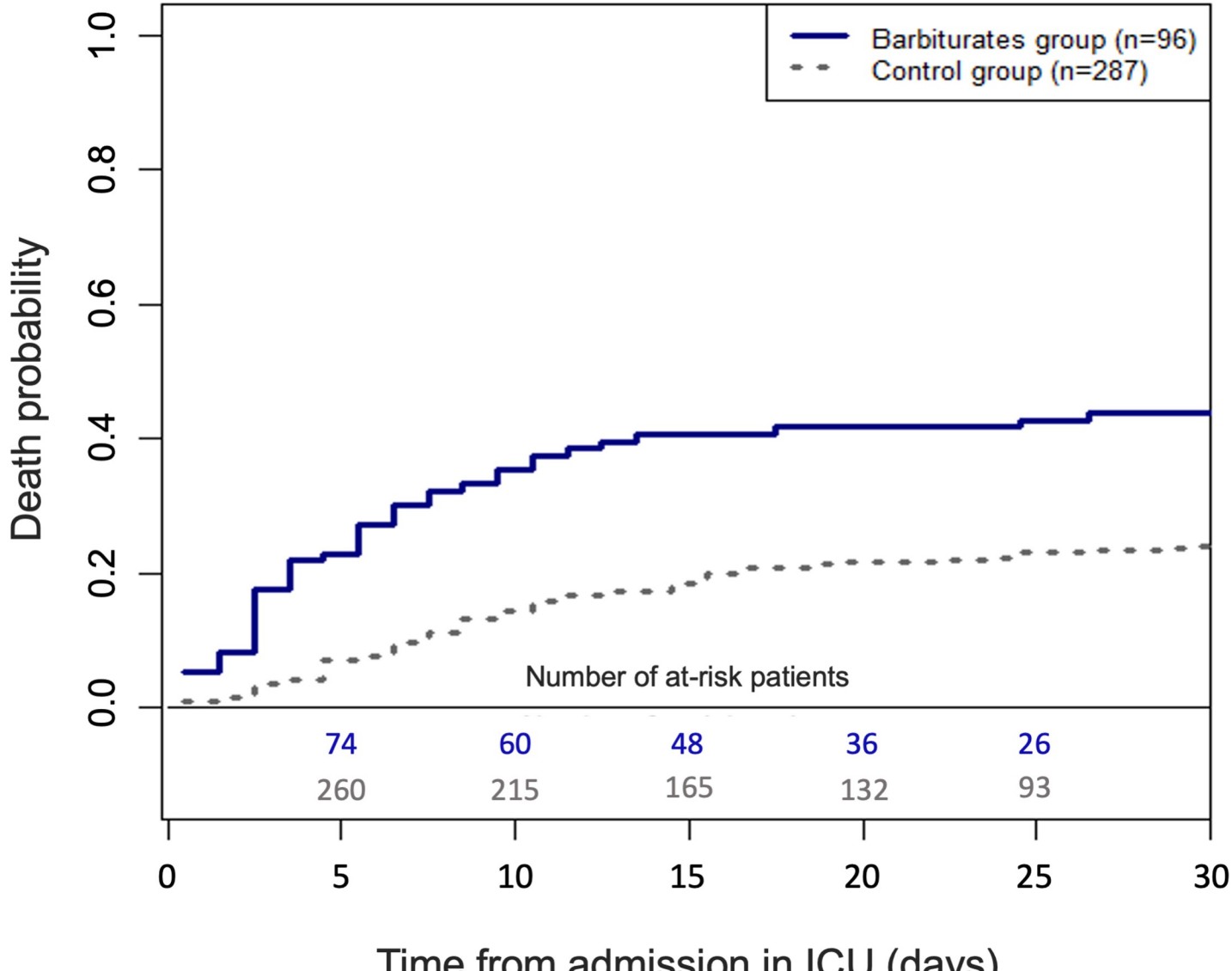

**Fig 2. Cumulative incidences curves for death in intensive care unit (estimated by using the Aalen-Johansen estimator with discharge as a competing event, n = 383).** The solid line represents the cumulative incidence curve for the barbiturates group (96 patients), while the dotted line corresponds to the control group (287 patients).

stay. In the barbiturates group, 49 (51.6%) patients developed a VAP compared to 158 (56.2%) in the control group. **Fig 3** shows the cumulative incidences of VAP. The observed cause-specific HR of VAP in ICU was 0.96 (95%CI from 0.69 to 1.32) for patients of the barbiturates versus the control group. The corresponding confounder-adjusted HR was 1.02 (95% CI from 0.75 to 1.41).

## GOS score at three months

The 3-month GOS was missing for 100 patients, who are excluded from this analysis. Characteristics at the time of admission in ICU of the included and excluded patients are presented in **S3 Table** in S1 File (supporting information). One of the centers was associated with a higher proportion of missing data on the GOS score.

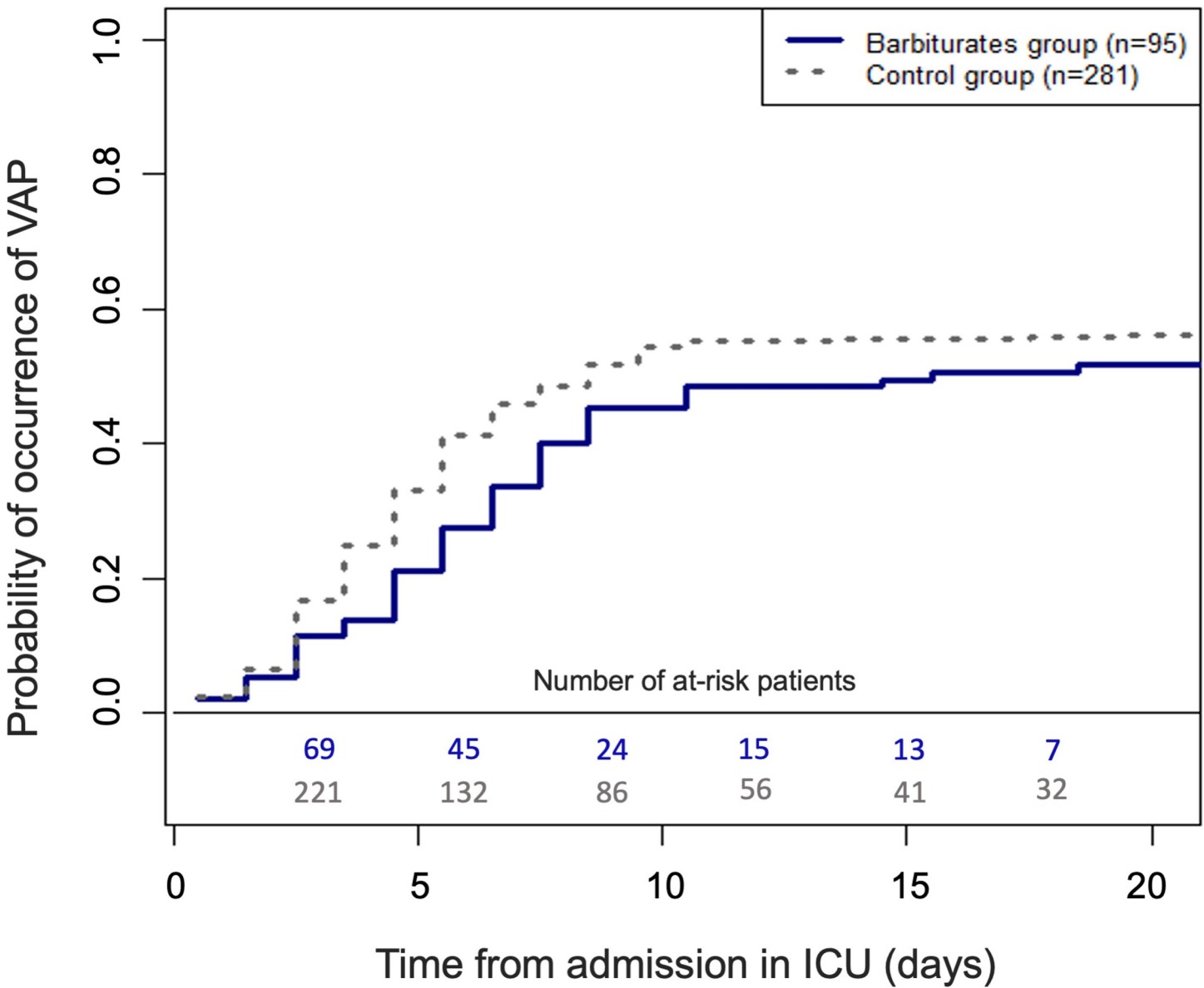

**Fig 3. Cumulative incidences curves for ventilator-associated pneumonia (VAP) in intensive care unit (estimated by using the Aalen-Johansen estimator with discharge and death as competing events, n = 376).** The solid line represents the cumulative incidence curve for the barbiturates group (95 patients), while the dotted line corresponds to the control group (291 patients).

The distribution of the 3-month GOS is presented in **Fig 4**. The overall percentage of patients with unfavorable prognosis was 66.1% (n = 187). The observed percentage was 74.7% in the barbiturates group versus 63.0% in the control group (p = 0.066). The corresponding confounder-adjusted percentages were 74.5% (95% CI from 61.2% to 84.4%) in the barbiturates group versus 63.7% (95% CI from 56.6% to 70.3%) in the control group. The corresponding OR of progressing to an unfavorable outcome at 3-months was 1.67 (95% CI from 0.84 to 3.33) for patients treated with barbiturates versus the others.

When we considered the propensity score model which included the IMPACT TBI score, the corresponding confounder-adjusted OR was 1.96 (95% CI from 0.93 to 4.17) for the barbiturates group versus the control group.

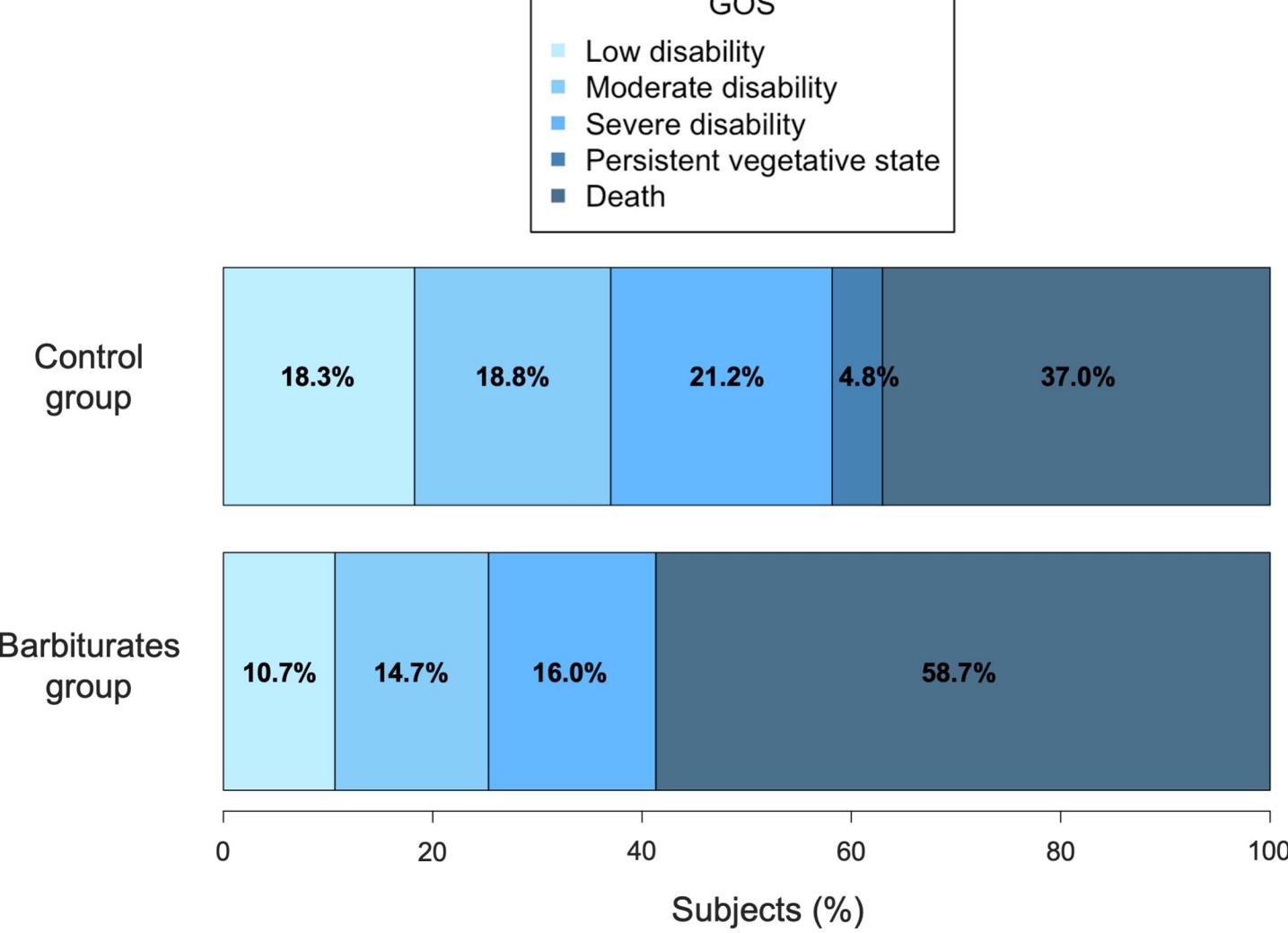

**Fig 4. Distribution of the 3-month Glasgow Outcome Scale score according to barbiturates treatment (n = 283).** The Glasgow outcome scale is represented from the light grey group (to the left of the bar plot) and corresponding to patients with little deficiency at three months, to the dark grey group (to the right of the bar plot) with a death status at three months.

### Sensitivity analysis

Among the 383 patients, 172 (44.9%) received barbiturates during their ICU stay at any time. During the follow-up, 117 patients died (76 (44%) among those who received barbiturates at any time and 41 (19%) among those who never received barbiturates). The cause-specific HR was 2.56 (95% CI from 1.72 to 3.70) for patients who received barbiturates at any time compared to those who did not. The cause-specific adjusted HR was 2.17 (95% CI from 1.35 to 3.45).

### Discussion

In our cohort of 1396 trauma patients, more than a quarter had early intracranial hypertension and around a quarter of these patients received early barbiturates (within 24-hours of admission). These patients had a lower ICU survival without a difference in the incidence of VAP or a in three-month functional outcome (i.e., the GOS at 3 months).

The first lesson of our cohort is that barbiturates appear to be frequently used to control early intracranial hypertension in TBI patients, with almost a quarter of them treated with barbiturates within 24 hours of admission. This proportion is in agreement with the one reported by Majdan et al. [14]. The question of the risk-benefit balance of this therapeutic is thus relevant [1]. Unlike Majdan et al. [14], we choose to select only patients with intracranial hypertension, since only this subgroup of patients may beneficiate from barbiturates, as a third-line therapy [1]. We defined intracranial hypertension as an ICP $\geq$ 20 mmHg, which this is the threshold of the French guidelines for therapeutic management [16]. We could have chosen a higher cut-off (22 or 25 mmHg), as proposed in the recent US guidelines [7]. But in France, treatment of elevated ICP is usually started for ICP $\geq$ 20 mmHg. We have also defined intracranial hypertension as the need for therapeutic intervention (i.e., before the measure of ICP), as it is often the case in real-life practices. Combining these criteria ensures us an exhaustive identification of the most severe patients eligible for barbiturates therapy within 24 hours of the intracranial hypertension episode. Despite these selection criteria, we still obtain a relatively large population (n = 383) compared, for example, to the most recent Cochrane review, in which 341 patients were studied (including 105 patients treated with barbiturates) [9]. There are indeed few trials evaluating barbiturates in severe TBI patients, and none are recent.

We reported a significant association between early barbiturates use and ICU survival. This finding is not in agreement with the result of the Cochrane review [9], and while all the precautions of causal inference have been considered, these findings emerged from an analysis of observational data (contrary to the Cochrane review which includes randomized trials). However, this may not be surprising, since barbiturates may impair cerebral oxygenation and thus may impair outcomes [26,27]. We must also underline that we focus on the early use of barbiturates (within the first 24 hours) when barbiturates are recommended as a third-line of therapeutic [1,7]. And even if our results highlight an adverse effect of the early use of barbiturates on ICU mortality, we cannot examine the later use of this therapeutic alternative in later onset episodes of intracranial hypertension.

However, we have reported that the long-term functional outcome is not significantly different, even if the GOS score tends to be worse in the barbiturates group. This result is in agreement with those obtained by Majdan et al. [14], who did not find any significant effect of barbiturates on the six-month GOS score. Few authors have studied the long-term outcomes of barbiturates-treated TBI patients. In 1985, Ward et al. have reported no significant difference in the one-year GOS score, from a randomized controlled trial of prophylactic pentobarbital versus standard treatment in 53 TBI patients [28]. Marshall et al. have suggested that barbiturates, used as rescue therapy in a cohort of 55 patients, was associated to a favorable functional outcome with 68% of survivors at one year, but without a control group [29]. In a series of mixed cases of 49 patients admitted for head injuries and subarachnoid hemorrhages, who have been treated with barbiturates, the results remained inconclusive on the GOS score at one year [30]. All these data are relatively old, and many practices have improved since. We have studied the outcome of patients treated according to recent recommendations and practices. In our study, the non-significant effect on the three-month GOS score should be interpreted in light of the number of missing data.

Barbiturates have been accused of being immunosuppressive and of promoting VAP. Indeed, the administration of barbiturates may promote reversible bone marrow suppression, inhibit normal monocyte behavior, and disrupt the NF-κB activating cascade [11,31,32]. In 1995, Nadal et al. have described a significant association of barbiturates uses and VAP occurrence in 151 patients with a head injury [12]. Lepelletier et al. have reported an adjusted OR of 2.68 (95% CI: 1.06–6.80) for the occurrence of early-onset ventilator associated-pneumonia during barbiturates therapy, in a cohort of 161 patients with head trauma [13]. The same

observation has been reported by Hamele et al. in a pediatric population [33]. However, in all these studies, the statistical adjustment may be insufficient. Contrary to these observations, our data do not allow us to conclude for a significant association on the occurrence of VAP. We should acknowledge that we only evaluate the association between early barbiturates use and the first episode of VAP, which may be partly related to pulmonary aspiration.

Our study has other limitations. Firstly, although we have analyzed a relatively large number of patients compared to the available literature, the lack of significant effect, particularly for the GOS score, can be attributed to a lack of power. The number of missing data for the 3-month GOS lowered the power of this analysis. Secondly, even if the variables collected at baseline seem to be able to describe the clinical picture of each patient, other confounders could have been omitted. That is why we also use the IMPACT TBI score in our propensity score Models. In the context of the analysis of observational data, no standardized barbiturates prescription protocol was used and cannot be reported. Physicians within the AtlanREA network follow US guidelines, using barbiturates as a second or third line [1,7]. Thirdly, the control group includes 76(19.8%) patients who have received barbiturates later in the course of their ICU stay. This highlights even more that barbiturates are frequently used in patients with intracranial hypertension (with a total of 172(45%) treated patients). The development of time-dependent propensity scores would have allowed to compare treated and untreated patients with the same characteristics at any post-admission time [34,35]. For that purpose, we need a cohort in which all the possible confounders are collected regularly (at least daily). Since we did not have all the data at each time points during the ICU stay, we restricted our purpose to the early use of barbiturates. However, we performed a sensitivity, exploratory analysis comparing the patients who received barbiturates at any time to those who did not and found consistent results, showing no benefit of barbiturates administration. In addition, we were not able to evaluate the dose and duration of barbiturates infusion, which could interfere with the effect of treatment [14]. The usual practice of the centers is to use high-doses barbiturates (boluses of 250–500 mg followed by continuous infusions around 4–8 g/24h) and to adjust them according to the measured ICP. Finally, it is a challenge to draw conclusions when all the factors leading the clinician to prescribe barbiturates cannot be controlled. Only a randomized controlled study may help to control these limits and to clarify the benefit of barbiturates in severe TBI patients.

## Conclusions

In our cohort of TBI patients with intracranial hypertension on admission, early use of barbiturates (within the first 24-hours) was associated with a lower ICU survival, but not to an increased incidence of VAP or poorer three-month functional outcome. In the absence of relevant clinical trials demonstrating the clinical benefit of barbiturates, each prescription requires a careful assessment of the benefit/risk ratio.

## Supporting information

**S1 File. Supplementary tables and figures.**
(DOCX)

**S1 Dataset. Anonymized dataset.**
(CSV)

## Acknowledgments

We thank the participating hospital centers of the AtlanREA Group: Angers, Nantes, Rennes, and Poitiers.

**AtlanRéa group consortium**

Soizic Gergaud,[1] Pierre Lemarié,[1]Thierry Bénard,[3] Rémy Bellier,[3] Dominique Demeure Dit-Latte,[4] Pierre-Joachim Mahe,[4] Yoann Launey,[5] Thomas Lebouvier,[5] Audrey Tawa,[5]

## Author Contributions

**Conceptualization:** Maxime Léger, Denis Frasca, Florent Le Borgne, Sigismond Lasocki.

**Data curation:** Maxime Léger, Denis Frasca, Antoine Roquilly, Philippe Seguin, Raphaël Cinotti, Claire Dahyot-Fizelier, Karim Asehnoune, Thomas Gaillard.

**Formal analysis:** Maxime Léger, Sigismond Lasocki.

**Investigation:** Maxime Léger, Raphaël Cinotti.

**Methodology:** Maxime Léger, Denis Frasca, Florent Le Borgne, Yohann Foucher, Sigismond Lasocki.

**Project administration:** Yohann Foucher, Sigismond Lasocki.

**Software:** Florent Le Borgne.

**Supervision:** Yohann Foucher, Sigismond Lasocki.

**Validation:** Antoine Roquilly, Philippe Seguin, Raphaël Cinotti, Claire Dahyot-Fizelier, Karim Asehnoune, Thomas Gaillard, Yohann Foucher, Sigismond Lasocki.

**Writing – original draft:** Maxime Léger, Florent Le Borgne, Thomas Gaillard.

**Writing – review & editing:** Antoine Roquilly, Philippe Seguin, Raphaël Cinotti, Claire Dahyot-Fizelier, Karim Asehnoune, Yohann Foucher, Sigismond Lasocki.

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
