## [Decision Letter · Decision Letter 0]

11 Jan 2021

PONE-D-20-37630

Early use of barbiturates is associated with increased mortality in traumatic brain injury patients—a propensity score-based analysis of a prospective cohort.

PLOS ONE

Dear Dr. Léger,

Thank you for submitting your manuscript to PLOS ONE. After careful consideration, we feel that it has merit but does not fully meet PLOS ONE’s publication criteria as it currently stands. Therefore, we invite you to submit a revised version of the manuscript that addresses the points raised during the review process.

Both reviewers have identified specific concerns with the analytical approach and a careful response and revision will be required.

We look forward to receiving your revised manuscript.

Kind regards,

Belinda J Gabbe, PhD

Academic Editor

PLOS ONE

Journal Requirements:

4. One of the noted authors is a group or consortium (AtlanREA). In addition to naming the author group, please list the individual authors and affiliations within this group in the acknowledgments section of your manuscript. Please also indicate clearly a lead author for this group along with a contact email address.

Reviewers' comments:

Reviewer's Responses to Questions

**Comments to the Author**

1. Is the manuscript technically sound, and do the data support the conclusions?

Reviewer #1: No

Reviewer #2: Partly

2. Has the statistical analysis been performed appropriately and rigorously? 

Reviewer #1: Yes

Reviewer #2: I Don't Know

3. Have the authors made all data underlying the findings in their manuscript fully available?

Reviewer #1: Yes

Reviewer #2: Yes

4. Is the manuscript presented in an intelligible fashion and written in standard English?

Reviewer #1: Yes

Reviewer #2: Yes

5. Review Comments to the Author

Reviewer #1: Thanks for asking me to review this study evaluating the association between barbiturate use and mortality in TBI patients. I have a number of comments:

1. The stated aim of this study is evaluate the effect of barbiturate therapy in patients with TBI and intracranial hypertension on outcomes including ICU survival; however, whether patients received barbiturate therapy or not is clearly subject to indication bias. Patients who received barbiturate therapy did so because the clinicians looking after them considered that this was indicated; presumably clinicians looking after the comparator patients did not give them barbiturates because they did not think barbiturates were indicated. Fundamentally, I do not think that it is valid to draw any conclusions about the effect of barbiturate therapy on outcomes based on these data. I do not consider that this problem is overcome in any meaningful way by the propensity score based analysis.

2. ICU mortality is a dichotomous variable and yet the primary analysis is presented as a time to event analysis with a hazard ratio. This analysis does not make sense from a clinical perspective. The relevant consideration is surely whether or not the patient dies in ICU rather than how long it takes for them to die?

3. As 3-month GOSE data are available in the study database, I would also presume that mortality data at 3 months must also be available. It is not clear to me why ICU mortality would be evaluated if mortality data were available at later time points. From the perspective of the patient, differences in ICU mortality rates that are not apparent by 3 months are probably not very important.

4. The proportion of patients with pneumonia is more clinically relevant that the time until 1st diagnosis of pneumonia and so, again, the analysis of pneumonia rates with the effect estimate presented as a hazard ratio does not really make clinical sense.

5. Findings limited to the subgroup of patients with confirmed elevated ICP are alluded to in the Discussion but do not appear to be included in the results.

6. There are some typographical and grammatical errors:

(i) "external ventricular derivations" (p4, line 70) should presumably be "external ventricular drainage" or similar

(ii) "wad" should be "was" (p6, line 98)

(iii) "All patients of the database..." (p6, line 105) should be "All patients in the database..."

(iv) "patient's comorbidities" (p7, line 121) should be "patients' comorbidities"

(v) "may beneficiate" (p14, line 253) should be "may benefit"

Reviewer #2: The authors present a propensity score-based analysis of a cohort of French TBI patients to determine the effect of early use of barbiturates on mortality, ventilated-associated pneumonia and functional outcome (GOS) at 3 months. Barbiturates are a commonly used therapy and understanding the risks and benefits is essential. The study has been well-conducted however I have some concerns which should be addressed to improve the quality of the manuscript.

1. Patents were reported as eligible if they had brain trauma, however no definition has been provided. Was a particular GCS required (e.g. <9) and were CT brain changes required? It is important for readers to understand the severity of TBI for included patients.

2. The barbiturates group was defined as patients receiving barbiturates in combination with other incremental therapies. It is assumed that this is barbiturates within the first 24 hours however this should be clearly stated. In addition, it should be clearly stated whether there was a minimum dose required or whether any administration of a barbiturate resulted in a patient being included in the barbiturates group. It is unclear why the authors chose barbiturates within 24 hours. Given they are recommended as a second or third tier therapy, many patients may receive barbiturates after 24 hours (after failure of other therapies). The discussion reports that 19.8% of control patients received barbiturates after 24 – a sensitivity analysis excluding these patients appears to have been conducted but not results are presented; this data should be included in the supplement.

3. It is unclear why a GCS <8 is reported rather than GCS<9 which is the commonly used definition for severe TBI. It is also of concern that GCS has not been used to determine the propensity weights. While the difference between the barbiturates group and control group is not significant, this may be due to the small sample size and a lack of power. Given the strong association between GCS and outcome, the authors should justify why it has not been included (taking into account the small sample size and likely reduced power to detect differences between groups).

4. The authors report that the CT severity of the injuries did not differ between the two groups. While there is not a statistically significant difference, the small sample size mean the power to detect a difference is low. There are 28.6% of patients with a category 1 or 2 Marshall score in the control group compared to only 17.7% in the barbiturates group.

5. There are other variables available that are known to be strongly predictive of outcome which have not been included in determining the propensity score. As with GCS above, the authors should provide information as to why pupil reactivity was not included (especially as there is a significant difference between groups).

6. The finding are not in agreement with the result of a Cochrane review of RCTs. The authors report this is not surprising, since barbiturates have been shown to impair cerebral oxygenation and may thus impair outcomes. This ignores the difference in study design and the authors should acknowledge the weaknesses of an observational design compared to RCTs.

7. The authors report that influential values were detected by a Cook distance greater than one in absolute value, however no information is provided about how many influential values there were. Given that observations with high weights can be unduly influential in a propensity score weighting, the authors should report how many values had a Cook distance greater than 1.

8. In Table S3, it is unclear if the proportion of missing data differed by group (barbiturates versus no barbiturates).

Minor concerns:

1. The financial disclosure statement is not complete and does not list the funder.

2. The results report that there were 1396 polytrauma patients however the discussion reports that there were 1396 TBI patients – please clarify which is correct.

6. PLOS authors have the option to publish the peer review history of their article (what does this mean?). If published, this will include your full peer review and any attached files.

Reviewer #1: No

Reviewer #2: No

---

## [Author Response · Author response to Decision Letter 0]

20 Dec 2021

We submit our manuscript as a new submission as we were in a reviewing process in Plos One on our previous manuscript (PONE-D-20-37630). We missed the response time. However, we did take into consideration the comments made by the reviewers and we offer a deeply revised version of our manuscript. We hope that we have addressed all the concerns and believe that these modifications have improved the manuscript substantially.

---

## [Decision Letter · Decision Letter 1]

9 Feb 2022

PONE-D-20-37630R1Early use of barbiturates is associated with increased mortality in traumatic brain injury patients—a propensity score-based analysis of a prospective cohort.PLOS ONE

Dear Dr. Léger,

Thank you for submitting your manuscript to PLOS ONE. After careful consideration, we feel that it has merit but does not fully meet PLOS ONE’s publication criteria as it currently stands. Therefore, we invite you to submit a revised version of the manuscript that addresses the points raised during the review process.

We look forward to receiving your revised manuscript.

Kind regards,

Aurel Popa-Wagner

Academic Editor

PLOS ONE

Journal Requirements:

Reviewers' comments:

Reviewer's Responses to Questions

**Comments to the Author**

1. If the authors have adequately addressed your comments raised in a previous round of review and you feel that this manuscript is now acceptable for publication, you may indicate that here to bypass the “Comments to the Author” section, enter your conflict of interest statement in the “Confidential to Editor” section, and submit your "Accept" recommendation.

Reviewer #2: (No Response)

2. Is the manuscript technically sound, and do the data support the conclusions?

Reviewer #2: Partly

3. Has the statistical analysis been performed appropriately and rigorously? 

Reviewer #2: Yes

4. Have the authors made all data underlying the findings in their manuscript fully available?

Reviewer #2: Yes

5. Is the manuscript presented in an intelligible fashion and written in standard English?

Reviewer #2: Yes

6. Review Comments to the Author

Reviewer #2: The reviewers have resubmitted their article which presents a propensity score-based analysis of a cohort of French TBI patients to determine the effect of early use of barbiturates on mortality, ventilated-associated pneumonia and functional outcome (GOS) at 3 months. The author have addressed many of my previous concerns and the analysis including the IMPACT-TCI score has strengthened the manuscript.

The following concerns should be addressed before publication:

1. The aim of the study is evaluate the effect of barbiturate therapy in patients with TBI and intracranial hypertension on outcomes including ICU survival. Patients who received early barbiturate therapy did so because clinicians believed there was an indication for barbiturates (and control patients did not receive barbiturates as clinicians presumably did not think barbiturates were indicated). It is challenging to draw conclusions about the effect of barbiturate therapy on outcomes where all factors leading to a clinician decision to prescribe barbiturates cannot be controlled for. This significant limitation should be clearly outlined in the discussion.

2. The authors also report that the findings are not in agreement with the result of a Cochrane review of RCTs. The authors report this is not surprising, since barbiturates have been shown to impair cerebral oxygenation and may thus impair outcomes. This ignores the difference in study design and the authors should acknowledge the weaknesses of an observational design compared to RCTs.

3. Please be consistent in the use of TBI patients (at times TBI patients is used and at times, brain-injured patients is used).

7. PLOS authors have the option to publish the peer review history of their article (what does this mean?). If published, this will include your full peer review and any attached files.

Reviewer #2: No

---

## [Author Response · Author response to Decision Letter 1]

14 Feb 2022

1. The aim of the study is evaluate the effect of barbiturate therapy in patients with TBI and intracranial hypertension on outcomes including ICU survival. Patients who received early barbiturate therapy did so because clinicians believed there was an indication for barbiturates (and control patients did not receive barbiturates as clinicians presumably did not think barbiturates were indicated). It is challenging to draw conclusions about the effect of barbiturate therapy on outcomes where all factors leading to a clinician decision to prescribe barbiturates cannot be controlled for. This significant limitation should be clearly outlined in the discussion.

We thank reviewer 2 for highlighting this point which is major in our analysis. In order to consider this proposal, we have ended our discussion with the following sentences:

“Finally, it is a challenge to draw conclusions when all the factors leading the clinician to prescribe barbiturates cannot be controlled. Only a randomized controlled study may help to control these limits and to clarify the benefit of barbiturates in severe TBI patients.”

2. The authors also report that the findings are not in agreement with the result of a Cochrane review of RCTs. The authors report this is not surprising, since barbiturates have been shown to impair cerebral oxygenation and may thus impair outcomes. This ignores the difference in study design and the authors should acknowledge the weaknesses of an observational design compared to RCTs.

The reviewer is right to point out that our analysis from observational data differs from the Cochrane review of randomized trials.

Our goal was to follow the rules of causal inference in order to reasonably emulate a pseudo-randomized trial (via the use of propensity scores). We are aware of the limitations of our analysis.

In order to clarify this point for readers, we have completed the end of our discussion by emphasizing the importance of conducting a randomized trial (see previous point):

“Finally, it is a challenge to draw conclusions when all the factors leading the clinician to prescribe barbiturates cannot be controlled. Only a randomized controlled study may help to control these limits and to clarify the benefit of barbiturates in severe TBI patients.”

In addition, we also propose to complete the sentence in the discussion referring to the Cochrane review as follows:

“This finding is not in agreement with the result of the Cochrane review [9], and while all the precautions of causal inference have been considered, these findings emerged from an analysis of observational data (contrary to the Cochrane review which includes randomized trials).”

3. Please be consistent in the use of TBI patients (at times TBI patients is used and at times, brain-injured patients is used).

As proposed, we have revised the manuscript to be consistent on the following term "TBI patients".

---

## [Editor Report · Decision Letter 2]

21 Apr 2022

Early use of barbiturates is associated with increased mortality in traumatic brain injury patients—a propensity score-based analysis of a prospective cohort.

PONE-D-20-37630R2

Dear Dr. Léger,

We’re pleased to inform you that your manuscript has been judged scientifically suitable for publication and will be formally accepted for publication once it meets all outstanding technical requirements.

Kind regards,

Aurel Popa-Wagner

Academic Editor

PLOS ONE
---

## [Editor Report · Acceptance letter]

26 Apr 2022

PONE-D-20-37630R2 

Early use of barbiturates is associated with increased mortality in traumatic brain injury patients from a propensity score-based analysis of a prospective cohort. 

Dear Dr. Léger:

I'm pleased to inform you that your manuscript has been deemed suitable for publication in PLOS ONE. Congratulations! Your manuscript is now with our production department. 

Kind regards, 

on behalf of

Professor Aurel Popa-Wagner 

Academic Editor

PLOS ONE